# Modification Strategy of D-leucine Residue Addition on a Novel Peptide from *Odorrana schmackeri*, with Enhanced Bioactivity and In Vivo Efficacy

**DOI:** 10.3390/toxins13090611

**Published:** 2021-08-31

**Authors:** Aifang Yao, Yingxue Ma, Xiaoling Chen, Mei Zhou, Xinping Xi, Chengbang Ma, Shen Ren, Tianbao Chen, Chris Shaw, Lei Wang

**Affiliations:** 1School of Pharmacy, Queen’s University Belfast, 97 Lisburn Road, Belfast BT9 7BL, UK; ayao01@qub.ac.uk (A.Y.); yma14@qub.ac.uk (Y.M.); x.xi@qub.ac.uk (X.X.); c.ma@qub.ac.uk (C.M.); t.chen@qub.ac.uk (T.C.); chris.shaw@qub.ac.uk (C.S.); l.wang@qub.ac.uk (L.W.); 2College of Chinese Medicinal Materials, Jilin Agricultural University, Changchun 130118, China; shenr@jlau.edu.cn

**Keywords:** brevinin, antimicrobial peptide (AMP), D-leucine peptide, cytotoxicity, dual antibacterial and anticancer activities

## Abstract

Brevinins are a well-characterised, frog-skin-derived, antimicrobial peptide (AMP) family, but their applications are limited by high cytotoxicity. In this study, a wild-type des-Leu2 brevinin peptide, named brevinin-1OS (B1OS), was identified from *Odorrana schmackeri*. To explore the significant role of the leucine residue at the second position, two variants, B1OS-L and B1OS-D-L, were designed by adding L-leucine and D-leucine residues at this site, respectively. The antibacterial and anticancer activities of B1OS-L and B1OS-D-L were around ten times stronger than the parent peptide. The activity of B1OS against the growth of Gram-positive bacteria was markedly enhanced after modification. Moreover, the leucine-modified products exerted in vivo therapeutic potential in an methicillin-resistant *Staphylococcus aureus* (MRSA)-infected waxworm model. Notably, the single substitution of D-leucine significantly increased the killing speed on lung cancer cells, where no viable H838 cells survived after 2 h of treatment with B1OS-D-L at 10 μM with low cytotoxicity on normal cells. Overall, our study suggested that the conserved leucine residue at the second position from the N-terminus is vital for optimising the dual antibacterial and anticancer activities of B1OS and proposed B1OS-D-L as an appealing therapeutic candidate for development.

## 1. Introduction

The *Enterococcus faecium*, *Staphylococcus aureus* (*S. aureus*), *Klebsiella pneumoniae*, *Acinetobacter baumannii*, *Pseudomonas aeruginosa*, and *Enterobacter species* (ESKAPE) are the main cause of hospital-acquired infections with limited treatment options [1]. Currently, drug-resistant infections associated with high mortality and economic burden tend to be some of the most serious threats to human health [2,3]. Another challenge is the rising incidence of cancer, which is the second leading cause of death in the world [4]. Furthermore, researchers have noted that patients suffering from a chronic infection with a weakened immune system are one of the leading causes of cancer [5]. Nowadays, commercially available drugs for the treatment of infections and cancers are facing two major problems, i.e., low selectivity and drug resistance. It is therefore urgent to explore new strategies against both of the above global health concerns. Fortunately, antimicrobial peptides (AMPs), a type of molecule characterised by cationicity and amphipathicity, present in amphibian skin secretion display broad-spectrum activity against bacteria and show significant anticancer activity, in addition to immune modulation [6]. In this context, some AMPs with dual antimicrobial and anticancer activities, along with a fast-killing pattern and lower toxicity, could be regarded as potential and promising therapeutic leads for clinical application overcoming desperate diseases [7,8].

Brevinin-1 peptides were first isolated from the skin of *Rana brevipoda porsa* [9]. Currently, researchers have explored more than 350 different AMPs from frog skin secretion belonging to the brevinin family and characterised by a typical conserved sequence in the cyclic heptapeptide domain (Cys-Lys-Xaa4-Cys), formed by a disulphide bond and named a “rana box” [10]. A previous review has also stated that the superfamily of brevinin peptides performs well in antibacterial activity and shows significant antiproliferative activity on human cancer cells. Even so, their strong haemolysis activity and cytotoxicity limit their application as antimicrobial agents [10]. Currently, the performance of AMPs can be improved by specific modification of natural peptide sequences, which is one of the current hotspots in research [11].

In this project, a wild-type des-Leu2 brevinin-1 peptide, namely brevinin-1OS (B1OS), was identified from the skin secretion of *Odorrana schmackeri* by “shotgun” cloning. Apart from the basic bioactivity evaluation of B1OS, we also focused on exploring the importance of the missing leucine residue at the second position in the parent peptide sequence. Hence, an α-helicity- and hydrophobicity-enhanced variant, B1OS-L, was designed by adding a leucine residue at position two of the parent sequence. Meanwhile, a D-leucine-modified peptide, B1OS-D-L, was designed to confirm the significance of this site. Peptides were chemically synthesised by solid-phase chemistry and purified using reverse-phase high-performance liquid chromatography (RP-HPLC) and matrix-assisted laser desorption/ionisation time-of-flight (MALDI-TOF) mass spectrometry. The in vitro antibacterial activity, anticancer activity, and general cytotoxicity were assessed. Additionally, the in vivo antibacterial activities of B1OS-L and B1OS-D-L were examined using an methicillin-resistant *Staphylococcus aureus* (MRSA)-infected insect larvae model.

## 2. Results

### 2.1. “Shotgun” Cloning of B1OS from Odorrana schmackeri Skin Secretion-Derived cDNA Library

A cDNA encoding the precursor of a novel peptide B1OS was cloned from the cDNA library constructed from *Odorrana schmackeri* skin secretion. The obtained nucleotide sequence was translated into an amino acid sequence (Figure 1A. Specifically, the characteristics of these highly conserved sequences include a putative signal peptide comprising 22 amino acid residues, an acidic spacer domain with Glu (E) and Asp (D), and a typical processing site for propeptide convertase (-KR-), which was located before a mature peptide. Additionally, a rana box was present at the C-terminal of the mature peptide that contained 23 amino acids. The sequences of B1OS (*Odorrana schmackeri*), brevinin-1SHb, brevinin-1HN1 and brevinin-1V (*Odorrana hainanensis*) [12], brevinin-1-RAB2, brevinin-1E-Sha, and brevinin-1HS [13] were aligned as shown in Figure 1B by the UniProt database (available online: https://www.uniprot.org/ (accessed on 10 March 2021)). This revealed that B1OS had highly conserved amino acids in common with the homologous peptide belonging to the brevinin-1 family. It is worth noticing that the hydrophobic leucine residue, which was highly conserved in the brevinin-1 family, was absent from the second position of the parent peptide B10S sequence. The nucleotide sequence of peptide B1OS was deposited in the GenBank database under Accession Number MZ450143. B1OS was subsequently isolated and identified from the skin secretion of *Odorrana schmackeri* (Appendix A).

### 2.2. Conformational Analysis of B1OS, B1OS-L and B1OS-D-L

The secondary structure of peptides was predicted by the employment of I-TASSER [14,15,16], and the parameter properties were evaluated by Heli-quest (Figure 2 and Table 1). According to the comparative alignment in Figure 1B, we assumed a conservative leucine was added at the second site based on the parent sequence and applied I-TASSER for structure prediction. These three peptides adopted a random coil conformation in the aqueous environment (10 mM ammonium acetate (NH_4_AC) solution) (Figure 3A), with a negative band at around 200 nm. As shown in Figure 3, the B1OS and its two analogues tended to adopt a structural transition with different degrees from random coil to helical structure in the presence of 50% of trifluoroethanol (TFE). Additionally, the result of the circular dichroism (CD) spectrum (Figure 3) was consistent with the prediction of the secondary structure. This indicated that B1OS-L tends to form a higher alpha-helix structure than B1OS in 50% TFE/10 mM NH_4_AC solution. Compared with B1OS-L, the addition of a D-leucine residue at the second position of the parent peptide sequence resulted in a slight decrease in helicity in 50% of TFE, but it was also higher than that of the parent peptide.

### 2.3. Minimum Inhibitory Concentration (MIC) and Minimum Bactericidal Concentration (MBC) of B1OS, B1OS-Lm and D-B1OS-L

The antimicrobial ability of B1OS and its analogues was evaluated using six microorganisms. The results of MIC and MBC assays are summarised in Table 2 and Appendix A. The original peptide, B1OS, displayed a moderate capacity against the growth of bacteria, with MICs ranging from 32 μM to 128 μM. Interestingly, the modified peptides B1OS-L and B1OS-D-L, designed by the addition of an L-leucine and D-leucine residue at position two, exerted a remarkable effect on antibacterial activity. In particular, the ability against the growth of Gram-positive bacteria was improved sharply after modification. The MICs against *S. aureus*, MRSA, and *Enterococcus faecalis* (*E. faecalis*) were enhanced from 32 μM to 2 μM, 64 μM to 4 μM, and 64 μM to 8 μM, respectively, in comparison with the parent peptide B10S.

### 2.4. Prevention and Eradication of Biofilm by B1OS, B1OS-L and B1OS-D-L

The antibiofilm activity of B1OS, B1OS-L and B1OS-D-L was assessed, and the minimum biofilm inhibitory concentration (MBIC) and minimum biofilm eradication concentration (MBEC) results are summarised in Table 3 and Appendix A. The parent peptide showed slight antibiofilm activity, with an MBIC of 64 μM against *S. aureus* and 128 μM against *E. coli*. Notably, the hydrophobicity-enhanced variant B1OS-L could inhibit the biofilm formation of Gram-positive bacteria in the concentration range of 4 μM to 16 μM, indicating a significantly improved biofilm prevention ability. Regarding the D-leucine-modified peptide B1OS-D-L, this produced a similar antibiofilm activity to B1OS-L.

### 2.5. Time-Killing Kinetics of Peptides against S. aureus, MRSA and E. coli

Based on the MIC results, the time-killing assay was conducted to assess the time span of different concentrations of B1OS, B1OS-L, and B1OS-D-L acting on *S. aureus* (NCTC 6538), MRSA (NCTC 12493), and *E. coli* (ATCC 8739). According to Figure 4, B1OS-L and B1OS-D-L had a similar killing pattern, which was more efficient compared to the parent peptide B1OS. Additionally, B1OS and its two analogues showed a concentration-dependent effect on bacteria. To be specific, 1× MICs of B1OS-L and B1OS-D-L could eliminate more than 4 × 10^5^ CFU/mL of *S. aureus* after 60 min incubation, and the latter showed a faster bactericidal effect, as complete killing of 5 × 10^5^ CFU/mL viable cells occurred in the first two hours. At 2× MICs, no surviving *S. aureus* could be counted after 30 min incubation with B1OS-L and B1OS-D-L, while B1OS killed all bacteria after 90 min. As for resistant-strain MRSA (Figure 4B), 2× MICs of B1OS-L and B1OS-D-L killed bacteria completely after 90 min. The killing curve of *E. coli* is shown in Figure 4C, and there were no surviving bacteria in the presence of 1× MICs of B1OS-L and B1OS-D-L after 60 min. After 30 min incubation with 2× MIC, the D-leucine-modified peptide demonstrated a bactericidal effect against *E.coli* with an averaged reduction in the viable cell count of 4.9 × 10^5^ CFU/mL. However, it took an extra 30 min for B1OS to remove all bacteria. In addition, the population of *E. coli* decreased remarkably 10–60 min after treatment with 2× MICs of peptides.

### 2.6. Membrane Permeability of MRSA by B1OS, B1OS-L and B1OS-D-L

The permeabilisation of B1OS and its analogues on MRSA (NCTC 12493) was measured by the employment of the SYTOX Green Nucleic Acid Stain (Figure 5). B1OS, B1OS-L, and B1OS-D-L induced membrane permeabilisation of MRSA in a concentration-dependent manner. At sub-MIC concentrations, B1OS-D-L could only induce 25% membrane disruption after 120 min incubation, while B1OS-L and B1OS had lower levels of membrane permeabilisation. A subsequent increase in peptide concentration (at 1× MICs) of these three peptides resulted in potent permeabilisation. At a concentration of 2× MIC, the percentage of membrane destruction increased rapidly in the presence of B1OS from 0 min to 60 min, following a constant permeability rate at approximately 75% over the next hour. Regarding B1OS-L and B1OS-D-L, the permeabilisation rate increased from about 30% to about 100% after 60 min treatment.

### 2.7. Treatment of MRSA-Infected Waxworms with B1OS-L and B1OS-D-L

In the in vitro study, B1OS-L and B1OS-D-L exhibited excellent antibacterial activity. Therefore, this attribute was further investigated using the MRSA (NCTC 12489)-infected larvae model. As shown in Figure 6, the mortality of the MRSA-infected waxworms was reduced significantly by treatment with B1OS-L and B1OS-D-L. Additionally, no death of larvae was observed up to a dose of 20 mg/kg, indicating that B1OS-L and B1OS-D-L were nontoxic in waxworms at 20 mg/kg. It was found that around 25% of larvae infected by MRSA survived at the dose of 20 mg/kg after 5 days.

### 2.8. Anticancer Proliferative Activity of B1OS, B1OS-L and B1OS-D-L

The antiproliferative activity of B1OS, B1OS-L and B1OS-D-L was evaluated on human cancer cell lines, including H838, PC-3, U251MG, MCF-7, and HCT116, at a range of concentrations from 10^−9^ M to 10^−4^ M (Figure 7). In summary, the parent peptide B1OS performed against the growth of cancer cells only at the concentration of 100 µM, which was far less potent than the modified products B1OS-L and B1OS-D-L. B1OS-L exhibited broad-spectrum anticancer activity with an IC_50_ value between 3.976 µM and 11.03 µM (Table 4), while the IC_50_ of B1OS-D-L ranged from 2.553 µM to 3.721 µM. Therefore, the antiproliferative activity of B1OS-D-L was most pronounced in human lung cancer cells.

### 2.9. Cytotoxicity of B1OS, B1OS-L and B1OS-D-L

The cytotoxicity of B1OS and B1OS-L was assessed on non-small-cell lung cancer cells, H838, and normal human microvascular endothelial cells, HMEC-1, in a trypan blue exclusion assay. As shown in Figure 8, the peptides B1OS-L and B1OS-D-L showed a fast-killing pattern on cancer cells. Specifically, there were no surviving cells in the presence of 10 μM of B1OS-D-L after 2 h, indicating its strong efficiency. About 80% of lung cancer cells showed damaged membranes after exposure to B1OS-L at the concentration of 10 μM. On the contrary, the percentage of mortality of normal human cells, HMEC-1, was far less, around 15% at a concentration of 10 μM of B1OS-L and B1OS-D-L. Regarding the parent peptide, B1OS failed to induce cell death on both cell lines.

### 2.10. Haemolytic Activity of B1OS, B1OS-L and B1OS-D-L

The haemolytic activity of B1OS, B1OS-L and B1OS-D-L was tested by the employment of a 2% horse erythrocyte suspension (Figure 9). B1OS caused no haemolysis up to 128 μM. After the introduction of a leucine residue, haemolytic activity was increased. The results also indicated that B1OS-L was most haemolytic with an HC_50_ of 29.92 µM, and D-leucine substitution decreased haemolytic activity to an HC_50_ value of 74.5 µM (Table 2). Notably, the D-leucine-modified peptide, B1OS-D-L, caused lower haemolysis in comparison with B1OS-L, which was less than 10% at its maximum MIC concentration (32 μM).

## 3. Discussion

In this study, B1OS, a novel peptide, was isolated from the skin secretion of *Odorrana schmackeri*. Homology analysis and functional prediction used the BLAST programme and UniProt database, which showed that this peptide belonged to the brevinin-1 family, which has multiple potential functions. Like other brevinin-1 peptides (Figure 1B), in addition to having the distinctive feature of a rana box, B1OS had the highly conserved residues of Ala9, which tended to contribute to helix formation [19], and Pro14, which induced a stable kink facilitating toroidal pore formation in bacterial membranes [20]. Interestingly, by alignment comparison, it was found that an FLP motif at the N-terminus was highly conserved in most brevinin-1 family members, whereas a hydrophobic leucine residue was lacking at the second position of B1OS and brevinin-1HSb. Structure–activity studies of brevinin-1E indicated that the artificial deletion of three amino acids at the N-terminus greatly reduced helical content and hydrophobicity, resulting in decreased haemolytic activity [21]. In this case, B1OS-L was designed by adding a leucine residue at the second position, and the secondary structure prediction indicated that it had higher helicity in the presence of 50% TFE, as well as hydrophobicity.

The results of CD spectroscopy were consistent with our predictions. It has been suggested that the tendency of AMP to form an amphiphilic α-helix in a simulated membrane environment plays a key role in the activity of membrane destruction [22]. As with most amphibian AMPs (e.g., dermaseptin and ranatuerin), the tested peptides conformed to a random coil in the aqueous while forming a helical structure with the presence of the secondary TFE [23,24,25,26]. It has been shown that TFE induced and stabilised the α-helical structure by perturbating hydrophobic interactions and promoting intramolecular hydrogen bond formation [27,28,29]. The structural transformation of AMPs in different environments is related to their amphipathic properties, that is, peptides own the relative proportion and position of hydrophilic hydrophobic domains [30]. Therefore, linear peptides generally do not have a specific structure in solution unless interacting with the cell membrane by folding into a helix and inserting into the membrane [31]. Previous studies have pointed out that hydrophobic amino acid residues are essential for helix formation and the integrity of the amphipathic structure [32], which is consistent with our research that the helicity of B1OS was significantly increased after the addition of leucine. As shown in the CD spectrum, B1OS-D-L maintained an α-helix conformation after a single substitution with the D-leucine residue, but the helicity of B1OS-D-L was relatively lower than that of B1OS-L. It is generally believed that D-amino acid substitution may locally interfere with peptide helix formation [33]. Pouny and colleagues reported that D-amino acid substitutions in pardaxin caused helix disorder and reduced haemolytic activity significantly [34], which is in accordance with our results.

According to previous studies, the evolution of AMPs is driven by positive selection [35,36,37]. Sequence variants derived from these AMPs can exhibit optimised function [37]. Based on previous studies, the brevinin-1 family possesses broad-spectrum activity against microorganisms, as well as strong haemolysis [10,38]. However, it is notable that B1OS had low haemolytic activity and moderate antibacterial activity. We speculate that amino acid deletion at the N-terminus of the B1OS might be related to biological evolution. In other words, the presence of a leucine residue at the second position is the result of gene mutations. The N-terminal FP motif evolved into FLP in brevinin-1 family peptides to adapt to the challenges of environmental changes. Therefore, stronger antibacterial ability and haemolytic activity were obtained.

To further confirm the significant role of a leucine residue at this site, B1OS-D-L was designed by substituting L-leucine with D-leucine. As shown in the CD spectrum, B1OS-D-L maintains an α-helix conformation after incorporating the D-leucine residue. It was found that the helicity of B1OS-D-L was relatively lower than that of B1OS-L, which can be attributed to the limited effect of a single D-amino acid on the α-helix structure. As a whole, B1OS-L and its diastereomers showed similarity in antibacterial activity, whereas the haemolysis activity of the D-leucine-modified peptide was reduced. According to a review paper, the diastereomers of AMPs obtained by incorporating unnatural amino acids showed better stability in vivo and in vitro [39]. Proteases cannot split the bond formed by D-amino acids in peptides due to the difference in side-chain spatial structure. The study of Heather and Feix clarified the correlation between the helix content of peptides and the toxicity to eukaryotic cells [40]. Similarly, the researchers also pointed out that the application of D-amino acid substitutions on the nonpolar side of the peptide to systematically regulate the helicity could enhance the antimicrobial activity and lower cytotoxicity to normal cells [41]. It can be interpreted as the D-type unnatural amino acids having different amino groups and positions, participating in hydrogen bonding and having different bulkiness on the side chain [42]. In terms of chemical and physical properties, there is no distinction between L-amino acid and its D-counterpart. Obviously, the incorporation of D-type amino acids is a strategy that allows the destruction of the secondary structure and maintains the net charge or hydrophobicity. In this study, the leucine residue at the second position of the parent peptide sequence is indispensable for obtaining the stronger biological activity of B1OS. The single substitution of D-leucine maintains the overall hydrophobicity and reduces the helicity to a certain extent. Therefore, the optimised antibacterial and anticancer activity could be maintained with an appropriately decreased haemolysis.

Several studies have found that the performance of AMPs can be improved by directional modification of structural parameters such as helicity, hydrophobicity, and net charge [33,43,44]. Generally, the property of hydrophobicity determines the extent to which AMPs can penetrate the lipid bilayer [45]. Peptides with higher hydrophobicity can penetrate further into the hydrophobic centre of the cell membrane, creating pores or channels on the membrane. The critical role of hydrophobicity in the antibacterial activity of KLA peptides was proven by Dathe and coworkers [46]. To some extent, increased hydrophobicity induced greater antibacterial and anticancer activity, as well as lower selectivity. For example, a membrane simulation model demonstrated that the higher hydrophobicity of G (ILKK) 3 L-NH_2_ provided stronger haemolysis activity [25]. Nevertheless, a previous study also pointed out that there was a threshold hydrophobicity at which better bioactivity could be achieved [47]. It is reasonable to suppose that the addition of a leucine residue made the overall hydrophobicity of peptides within the optimal hydrophobicity window, resulting in enhanced antibacterial and anticancer activity. To some extent, the increased percentage of a-helicity also contributed.

The antimicrobial assays showed that B1OS-L and B1OS-D-L were far more potent in inhibiting the growth of microorganisms than the parent peptide, with MICs ranging from 2 μM to 8 μM against Gram-positive bacteria. Additionally, the results also revealed that the susceptibility of Gram-positive bacteria was more than that of Gram-negative bacteria. This phenomenon is consistent for most peptides of the brevinin-1 family [12]. Normally, the reason can probably be explained by the difference in membrane structures and membrane compositions. In particular, Gram-positive bacteria have the negative charge of teichoic acids (WTAs) and lipoteichoic acids (LTAs) that are attached to the peptidoglycan [48], while Gram-negative bacteria have an extra outer membrane composed of negatively charged lipopolysaccharide (LPS), making their permeabilisation more difficult [48].

Further, the results of time-killing and membrane permeability assays showed that peptides acted in a concentration-dependent pattern. Accordingly, a limited number and size of toroidal pores was probably formed at lower concentrations, whereas a higher density of peptides can accumulate on the lipid membrane acting through the carpet model, leading to membrane lysis [49]. Hartmann et al. used SEM and TEM images and proposed that AMPs could only induce membrane leakage at a low concentration, while higher concentrations (>MIC) could cause membrane disintegration and cell lysis [50]. Researchers also proposed that 1xMIC of PSN-PC could not cover the entire lipid layer; however, this peptide was reported to tend to act in a carpet and toroidal pore mode under the concentrations of 2× MIC and 4× MIC [51]. In this case, B1OS, B1OS-L, and B1OS-D-L were likely to act in the same way. 

B1OS-D-L showed potent antimicrobial activity in comparison to other peptides from the brevinin-1 family. For example, the MIC of brevinin-1SPb and brevinin-1DYa against the growth of *S. aureus* was 13 µM and 32 µM with strong haemolysis [52]. On the other hand, by comparison with other brevinin-1 peptides, such as Ose [53], brevinin-1H [54], and B1A [55], B1OS-D-L showed comparable or even better antibacterial activity against Gram-positive bacteria with a lower level of haemolysis. Researchers have shown that the immune response of waxworms is quite similar to that of mammals [56]. Regarding the promising antibacterial activity of B1OS-L and B1OS-D-L, in vivo efficacy was assessed in an MRSA-infected *Galleria mellonella* larvae model [57]. It was found that B10S-L and B1OS-D-L could reduce the mortality rate of infected waxworms but were less potent compared with current antibiotics, which might be attributed to their poor stability in vivo. Generally, peptides are unstable under serum conditions and are easily degraded by proteases, leading to loss of activity or poor bioavailability [58]. It was also found that B1OS-D-L caused low haemolysis at the maximum active concentration (32 µM), and it induced no death of larvae, which supplied a safety reference for further study.

Apart from the excellent antibacterial activity of AMPs, their potential value as cancer therapeutics has attracted the attention of researchers [59]. Here, indeed, the antiproliferative assay and trypan blue exclusion assay indicated that B1OS-L and B1OS-D-L performed potently against the growth of cancer cells. Attractively, leucine-modified peptides, especially B1OS-D-L, performed fast killing against lung cancer cells within two hours. Meanwhile, it showed low cytotoxicity to normal human cells. Generally, the distinctions of membrane composition between cancer cells and normal mammalian cells accounted for the selectivity and activity of AMPs. Concerning cancer cells, they typically possess negative charges because of the elevated number of anionic molecules such as phosphatidylserine, O-glycosidic mucins, and sialylated gangliosides, which could facilitate AMP interaction [60].

## 4. Materials and Methods

### 4.1. Acquisition of Odorrana schmackeri Skin Secretion

Specimens of the frog, *Odorrana schmackeri* (*n* = 3), were captured in Fujian Province, People’s Republic of China. All the frogs were adults, and the skin secretions were acquired by the employment of mild transdermal electrical stimulation on dorsal skin. Subsequently, the stimulated viscous secretions were washed by deionised water and were snap-frozen with liquid nitrogen before lyophilisation. The lyophilised skin secretions were stored at −20 °C prior to mRNA isolation. The procedure of obtaining skin secretion was administered by the Institutional Animal Care and Use Committee (IACUC) of Queen’s University Belfast and approved on 1 March, 2011. It was performed with the accordance of UK Animal (Scientific Procedures) ACT 1986, Project license PPL 2694, issued by the Department of Health, Social Service and Public Safety, Northern Ireland.

### 4.2. “Shotgun” Cloning of B1OS from Odorrana schmackeri Skin Secretion-Derived cDNA Library

Five milligrams of lyophilised skin secretion was dissolved by adding 1 mL of lysis/binding buffer. mRNA was released by Dynabeads® mRNA DIRECT™ Kit (Dynal Biotech, UK) based on the principle of covalent pairing. The isolated mRNA was regarded as the template guiding the first-strand synthesis for cDNA library construction by a SMART-RACE Kit (Clontech, UK) with a nested universal primer (NUP) and degenerate sense primer (S: 5′-CARAAYACITAYMGIGCICC-3′; R = A + G, Y = C + T, M = A + C). The cDNA ends were rapidly amplified by PCR system, which began with the initial denaturation at 94 °C and 35 cycles, including 30 s denaturation at 94 °C, primer annealing at 60 °C for 30 s, 72 °C extension for 3 min, and finished with a final extension at 72 °C for 10 min to purify PCR products by a Cycle Pure Kit (Omega Bio-Tek, Norcross, GA, USA). The purified products were amplified using the pGEM-T Easy Vector (Promega, Madison, WI, USA) and analysed by gel electrophoresis. The sequence of nucleotides in the target DNA was acquired by the process of DNA sequencing achieved by Big Dye Terminator v. 3.1 Cycle Sequencing Kit (Applied Biosystems, Foster City, CA, USA), along with an ABI3730 automated sequencer (Life Technologies, Paisley, UK). To further confirm whether this transcript was actually expressed, a further 5 mg of frog skin secretion dissolved in 1 mL of 0.05/99.5 (v/v) trifluoroacetic acid (TFA)/water) was injected into an RP-HPLC column for fractionation. The collected fractions were then subjected to primary structural analysis by MS/MS fragmentation. Full details of elucidation of peptides from *Odorrana schmackeri* skin secretion were described elsewhere [38].

### 4.3. Synthesis and Purification of B1OS, B10S-L and B1OS-D-L

Solid-phase peptide synthesis was employed to obtain sufficient peptides for bioactivity analysis. Peptides were chemically synthesised according to the mature peptide sequence of translated open-reading fragments in molecular cloning by a solid-phase peptide synthesiser (Tribute 2-Channel Peptide Synthesiser, Protein Technologies, Tucson, AZ, USA) along with Fmoc-Cys(Trt)-Wang resin (100–200 mesh) (Novabiochem, Germany) and standard Fmoc chemistry. The amino acids were weighed and mixed with 2-(1H-benzotriazole-1-yl)-1,1,3,3-tetramethyluronium hexafluorophosphate (HBTU). The synthesis system used four solutions, including dimethylformamide (DMF) (Sigma-Aldrich, 99%), 20% piperidine (Sigma-Aldrich, 99%), 11% N-methylmorpholine (NMF) (Sigma-Aldrich, 99%), and 100% dichloromethane (DCM) (Sigma-Aldrich, 99%). The mixture of cleavage solution (25 mL/1 g) that contained 94% TFA, 2% water, 2% TIS (triisopropylsilane), and 2% EDT (1,2-ethanedithiol) was added at room temperature with stirring for 2 h to remove peptides from the solid support and the side-chain-protecting groups. Cold diethyl ether was added to precipitate peptides from the cleavage solution. The precipitated peptide was then washed by cold diethyl ether and eventually dissolved by 5 mL of acetonitrile/H_2_O/TFA (80/19.95/0.05, v/v/v) and 15 mL of H_2_O/TFA (99.95/0.05, *v*/*v*) before lyophilisation. The lyophilised samples were purified using a Cecil CE4200 Adept (Cambridge, UK) RP-HPLC and identified by MALDI-TOF mass spectrometry (Perspective Biosystems, Foster city, CA, USA) (Appendix A).

### 4.4. Prediction of Peptide Structure

The secondary structure of a series of derivatives of B1OS was predicted using I-TASSER (available online: https://zhanglab.ccmb.med.umich.edu/I-TASSER/about.html, accessed on 19 June 2021), and the structural parameters were analysed by Heli-quest (available online: https://heliquest.ipmc.cnrs.fr/cgi-bin/ComputParams.py, accessed on 19 June 2021). The sequence of B1OS, B1OS-L, and B1OS-D-L was submitted to the online platform of I-TASSER, and the outputs of modelling results were summarised in a web page by sending an email link. The structural parameters were obtained by the process of submitting the sequence of B1OS, B1OS-L, and B1OS-D-L to Heli-quest.

### 4.5. CD Spectra

The secondary structures of B1OS, B1OS-L, and B1OS-D-L were investigated using a CD spectrometer (JASCO-815). The peptide was dissolved in 10 mM NH_4_AC buffer and 50% TFE (v/v in 10 mM NH_4_AC), respectively, with a final concentration of 100 μM, and then was loaded into a 1 mm path length cuvette. The sample solution was analysed within the range of 190 to 250 nm at room temperature at a scanning speed of 100 nm/min, a bandwidth of 1 nm, and a data pitch of 1 nm.

### 4.6. MIC and MBC Assays

The MIC and MBC assays were used to conduct the inhibitory and bactericidal abilities of B1OS and its designed variants on the growth of planktonic microorganisms. Six microorganisms were used to detect the antimicrobial activity of peptides, including Gram-positive bacteria *S. aureus* (NCTC 6538), MRSA (NCTC 12493), *E. faecalis* (NCTC 12697), Gram-negative bacteria *E. coli* (ATCC 8379), *K. pneumoniae* (ATCC 43816), and *P. aeruginosa* (ATCC 9027). The microorganisms were inoculated in Mueller–Hinton Broth (MHB) in a shaking incubator at 37 °C overnight, and the bacteria were diluted to 5 × 10^5^ CFU/mL after reaching the logarithmic growth phase. Peptides were dissolved in DMSO to make the concentration of 512 × 10^2^μM as a stock solution, and then each stock solution was two-fold diluted to obtain a concentration from 256 × 10^2^ μM to 100 μM. One microlitre of each concentration of peptide solution was added to a 96-well plate and incubated with 99 μL of diluted bacteria for 18 h at 37 °C. The absorbance of each well was determined at a wavelength of 550 nm using the Synergy HT plate reader (Bio-Tek, Minneapolis, MN, USA). As for those wells that were clearly visible, 20 μL of solutions was dropped on an MHA dish and then incubated at 37 °C overnight to determine the MBC.

### 4.7. MBIC and MBEC Assays

The MBIC and MBEC assays were used to investigate the ability of B1OS and its analogues to inhibit and eradicate the formation of biofilm. The microorganisms were used as described above. The bacteria were inoculated and diluted to 5 × 10^5^ CFU/mL after arriving at the logarithmic growth period. In the MBIC assay, 1 μL of each concentration of the peptide solution (ranging from 512 × 10^2^ μM to 100 μM) was added into a 96-well plate and incubated with 99 μL of diluted bacteria for 16–20 h at 37 °C. As for the MBEC assay, 100 μL of cultured bacteria was seeded in a 96-well plate, and the plate was placed on a shaking incubator at 37 °C for 24 h. Subsequently, each well was rinsed twice with 100 μL of sterile PBS to remove planktonic cells and treated with 100 μL of each concentration of peptides (ranging from 1 μM to 512 μM). After 24 h incubation, the plate was washed twice by 100 μL of PBS and stained with the same volume of 0.1% crystal violet solution (Sigma-Aldrich, Overijse, Belgium). After this, 100 μL of 30% of acetic acid (Sigma-Aldrich, UK) was transferred to each well for dissolution. The absorbance was monitored by use of a Synergy HT plate reader at 595 nm.

### 4.8. Time-Killing Assays

The time-killing kinetic assay was used to determine the kinetic killing of bacteria by various concentrations of peptides. *S. aureus* (NCTC 6538), MRSA (NCTC 12493), and *E. coli* (ATCC 8739), which were the common and representative strains in clinical practice, were selected. The bacteria were inoculated in MHB in a shaking incubator at 37 °C overnight. The bacteria were diluted to 5 × 10^5^ CFU/mL after reaching the logarithmic growth phase. Two hundred microlitres of diluted bacteria was added to a 1.5 mL tube as the growth control. Then, 2 μL of 2× MIC, 1× MIC of B1OS, B1OS-L and B1OS-D-L was mixed with 198 μL of diluted bacteria as experimental groups. All the tubes were incubated for 0, 10, 20, 30, 60, 90, 120, and 180 min. At each time point, 10 μL of each solution was added to 90 μL of PBS to make a 10^1^ times dilution, and 10 μL of the mixture was added to 90 μL of PBS. This step was repeated six times to obtain 10^6^ dilutions for the viable count. Next, 10 μL of different diluted solutions were dropped on the MHA plate. All agar plates were incubated at 37 °C overnight, and the number of colonies was calculated. The CFU/mL could be calculated using the formula, CFU/mL = (Number of colonies × dilution factor) / volume of culture.

### 4.9. Bacterial Membrane Permeability Assays

The effect of B1OS, B1OS-L, and B1OS-D-L on the membrane integrity of MRSA (NCTC 12493) was detected by the application of SYTOX Green Nucleic Acid Stain (Thermo Fisher Scientific, Waltham, MA, USA). MRSA was inoculated in TSB at 37 °C overnight and subculture in 25 mL of TSB to arrive at the logarithmic growth phase in a shaking incubator at 37 °C for 2.5 h. The supernatant was poured out after centrifugation (1000× *g*, 10 min, 4 °C), and then 30 mL of 5% TSB in 0.85% NaCl solution was added to resuspend the bacterial cell pellet. This washing step was repeated twice, and the bacteria were suspended until about a 1 × 10^8^ CFU/mL density by measuring the OD value (0.7) at a wavelength of 590 nm. Forty microlitres of B1OS, B1OS-L, and B1OS-D-L at the concentration of 2× MIC was added in a black 96-well plate. Meanwhile, 10 μL of diluted SYTOX Green Nucleic Acid Stain (5 μM) was transferred into each well. The 100% membrane permeabilisation was achieved by treating the bacteria with 70% isopropanol, and the bacterial suspension with 5% TSB in 0.85% NaCl solution was regarded as the growth control. The plate was immediately placed in the Synergy HT plate reader by applying a 2 h kinetic programme with excitation at 485 nm and emission at 580 nm after adding 50 μL of a bacterial suspension at 1 × 10^8^ CFU/mL to each well by a multichannel pipette. The percentage of permeability can be calculated by comparing the fluorescent density of bacterial suspension treated with 70% isopropanol and 5% TSB in 0.85% NaCl.

### 4.10. Efficacy Evaluation of B1OS-L against MRSA in Larvae

The in vivo antibacterial activity of B1OS-L and B1OS-D-L was assessed using the larva of *Galleria mellonella* according to a previous publication with minor revisions [61]. The waxworms (250 ± 50 mg) (Livefood UK Ltd., Rooks Bridge, UK) were injected with 10 μL of MRSA (NCTC12493) suspension (5 × 10^7^ CFU/mL), which was prepared in sterilised PBS. After 1 h of infection, each waxworm was administered with 10 μL of 10 mg/kg and 20 mg/kg of B1OS-L and B1OS-D-L. The same injection volumes of PBS and 20 mg/kg vancomycin were regarded as negative control and positive control. Each group contained ten waxworms, and the number of survival and death was observed every 12 h for five days.

### 4.11. Anticancer Proliferative MTT Assays

Five human cancer cell lines—non-small-cell lung cancer H838, human prostate carcinoma cell line PC-3, human glioblastoma astrocytoma U251MG, human breast cancer cell line MCF-7, and human colorectal carcinoma cell line HCT116—were used to detect the anticancer proliferative ability of B1OS and its analogues based on the application of MTT, which can measure cell viability by reducing yellow MTT to purple formazan. H838 and PC-3 were cultured in Roswell Park Memorial Institute (RPMI)-1640 medium (Life Technologies, Paisley, UK) mixed with 10% (*v*/*v*) fetal bovine serum (FBS) and 1% penicillin–streptomycin (Gibco, 10,000 U/mL). U251MG and MCF-7 were cultured in Dulbecco’s Modified Eagle medium (DMEM) (Life Technologies, Carlsbad, CA) mixed with 10% (*v*/*v*) FBS and 1% penicillin–streptomycin (Gibco, 10,000 U/mL). HCT116 was cultured in McCoy’s 5A (modified) medium (Life Technologies, Carlsbad, CA) mixed with 10% (*v*/*v*) FBS and 1% penicillin–streptomycin (Gibco, 10,000 U/mL). Cells were seeded in 96-well plates at the density of 8000 cells per well and incubated at 37 °C with 5% CO_2_ overnight. The cells were treated with 100 μL of the serum-free medium for at least 4 h after attaching. The peptide solutions were prepared at a concentration of 10^−4^ M to 10^−9^ M in serum-free medium, 100 μL of each solution were transferred to each well, and the plate was incubated at 37 °C for 24 h. Triton X-100 (0.1%) served as the positive control, and the growth control was treated with the serum-free medium. Subsequently, 10 μL of 5 mg/mL 3-(4,5-dimethylthiazol-2-yl-)-2, 5-diphenyltetrazolium bromide (MTT) was added to each well under dark condition and incubated at 37 °C for 2 h. After that, the mixture was discarded and replaced with 100 μL of DMSO. The plate was placed on a shaking incubator for 5–10 min, and the absorbance was detected at a wavelength of 570 nm by the Synergy HT plate reader (Bio-Tek, Minneapolis, MN, USA).

### 4.12. Trypan Blue Exclusion Assay

The cytotoxicity of B1OS, B1OS-L, and B1OS-D-L was determined by trypan blue exclusion assay, which is based on the principle that live cells with intact cell membranes can exclude dyes, whereas dead cells cannot. H838 was cultured in RPMI-1640 medium (Life Technologies, Carlsbad, CA, USA) mixed with 10% (*v*/*v*) fetal bovine serum (FBS) and 1% penicillin–streptomycin (Gibco, 10000 U/mL). Normal human microvessel endothelial cell HMEC-1 was cultured in MCDB131 medium (Life Technologies, Carlsbad, CA, USA) mixed with 10% (*v*/*v*) FBS, 1% penicillin–streptomycin (Gibco, 10,000 U/mL), 10 ng/mL epidermal growth factor (EGF), 1 µg/mL hydrocortisone, and 10 mM glutamine. Cells were seeded into a 12-well plate at the density of 2 × 10^5^ cells per well at 37 °C with 5% CO_2_ overnight. After 4 h starvation, 500 μL of peptide solution in serum-free medium was added to each well to achieve the final concentrations of 10 μM and 5 μM. Cells were incubated at 37 °C for 2 h, 6 h, and 24 h. Cells with the presence of serum-free medium were treated as the growth control. At each time point, the solution which might contain dead cells in each well was collected in a 1.5 mL tube separately, and then 100 μL of trypsin (0.5%, Gibco, UK) was transferred into each well and incubated at 37 °C for 1–2 min, followed by adding 400 μL of complete medium to stop trypsinisation. The cell suspension was transferred to the 1.5 mL tube and then centrifuged (380× *g*, 5 min, 18 °C). After that, the supernatant was removed, and the cell pellet was resuspended by gently pipetting after adding 400 μL of PBS. Ten microlitres of cell suspension was mixed with the equivalent volume of trypan blue (0.4%, Gibco, St. Louis, MO, USA). Then, 10 μL of the mixture was loaded into a haemocytometer to perform cell counting. The percentage of viable cells and dead cells was calculated by counting the unstained and stained cells separately.

### 4.13. Haemolysis Assay

The haemolysis activity of B1OS and its designed peptides on mammalian erythrocytes was evaluated using 2% defibrinated horse blood (TCS Biosciences Ltd., Buckinghamshire, UK). Two millilitres of fresh horse blood was transferred into a 50 mL tube, and then 25–30 mL of sterilised PBS was added gently to wash the blood. The tube was placed gently on a shaker to mix sufficiently, and the supernatant was discarded after centrifuging at 930× *g* for 5 min. This step was repeated until the supernatant was visibly clear. After the washing step, the red blood cell pellet was diluted with 50 mL of PBS to achieve a 4% erythrocytes suspension. The concentration of peptide solution was prepared from 256 μM to 2 μM by two-fold dilution in sterilised PBS. One hundred microlitres of red blood cell suspension was added into a 1.5 mL tube treated with the same volume of each concentration of peptide and incubated at 37 °C for 2 h. The red blood cell suspension treated with 0.1% Triton X-100 and PBS was regarded as positive control and negative control. After centrifugation at 930× *g* for 10 min, 100 μL of suspension was transferred to a 96-well plate, and the absorbance was obtained by Synergy HT plate reader at 490 nm. The haemolysis rate can be calculated using the formula, haemolysis ratio = (absorbance (experimental groups) − absorbance (negative control))/ (absorbance (positive control) − absorbance (negative control)) × 100%.

## 5. Conclusions

In conclusion, we successfully designed a B1OS derivative with optimised dual antibacterial and anticancer activity with low toxicity by introducing a leucine residue at the second position of the parent sequence. Moreover, B1OS-D-L, a product obtained by incorporating a D-leucine in this site, behaved with comparable functions to B1OS-L with lower haemolytic activity. The D-leucine-modified peptide could kill drug-resistant bacteria MRSA and lung cancer cells without causing haemolysis and cytotoxicity. In addition, B1OS-D-L exerted potent in vivo antibacterial efficacy in an MRSA-infected larvae model. Thus, this study showed the significant role of the leucine residue in brevinin-1 peptides and that the modified peptide, B1OS-D-L, could be further analysed and explored as a promising agent to deal with drug resistance.

## Figures and Tables

**Figure 1 toxins-13-00611-f001:**
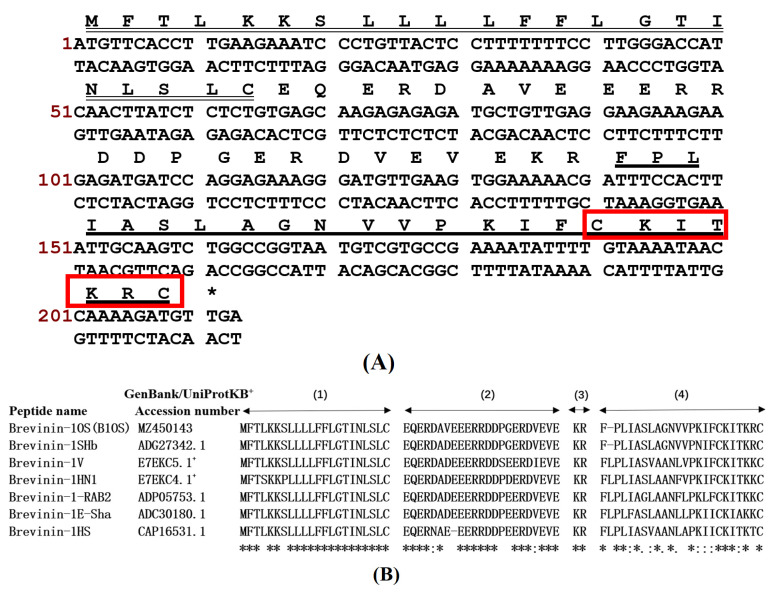
(**A**) Nucleotide sequence and the translation of open-reading-frame amino acids of the full length of a cDNA precursor from the skin secretion of *Odorrana schmackeri*. The putative peptide is double underlined, and the mature peptide is single underlined. The asterisk means stop codon, and the “Rana box” is highlighted by a red box. (**B**) Alignments of translated amino acid sequences of B1OS and brevinin family peptides derived from different frog species. The domains are divided into four parts, including the putative signal peptide (1), acidic spacer region (2), convertase processing site (-KR-) (3), and the mature brevinin peptide (4). The asterisks indicate conserved amino acid residues. The colon indicates strongly similar properties of conservation between groups. The period indicates weakly similar properties of conservation between groups.

**Figure 2 toxins-13-00611-f002:**
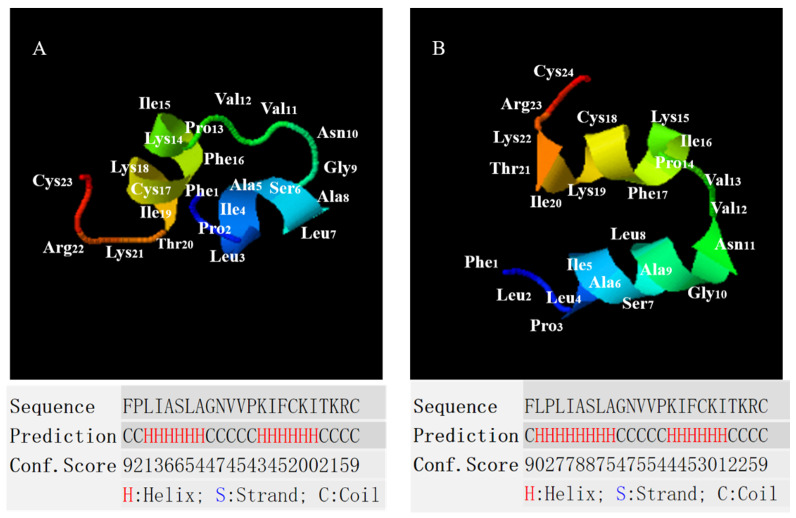
Secondary structure prediction of (**A**) B1OS and (**B**) B1OS-L using I-TASSER. The higher conf. score means the higher confidence of secondary structure prediction. Each amino acid and its position in the models are represented by a three-letter abbreviation and number, respectively.

**Figure 3 toxins-13-00611-f003:**
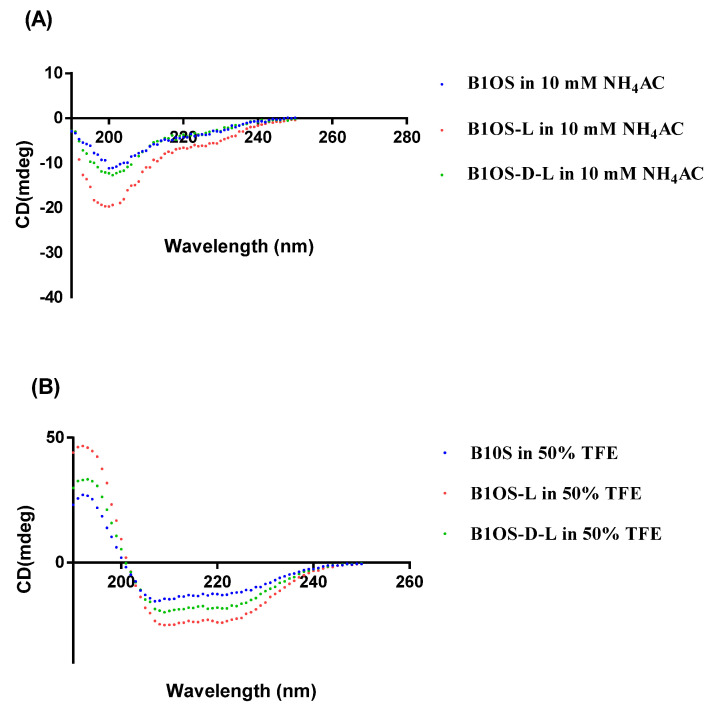
CD spectra of B1OS (blue), B1OS-L (red), and B1OS-D-L (green) in 10 mM NH_4_AC solution (**A**) and 50% TFE/10 mM NH_4_AC solution (**B**).

**Figure 4 toxins-13-00611-f004:**
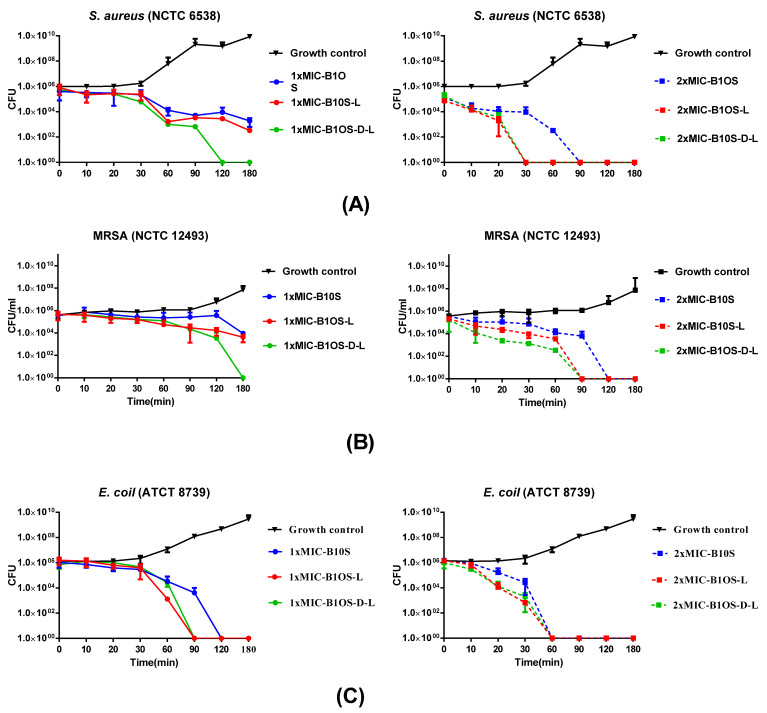
The kinetic time-killing curves of B1OS (blue), B1OS-L (red), and B1OS-D-L (green) against (**A**) *S. aureus*, (**B**) MRSA, and (**C**) *E. coli* at concentrations of 1-fold and 2-fold MICs. The peptide was added at 0 min and monitored until 180 min. Data were represented as the means ± standard deviation of three independent experiments.

**Figure 5 toxins-13-00611-f005:**
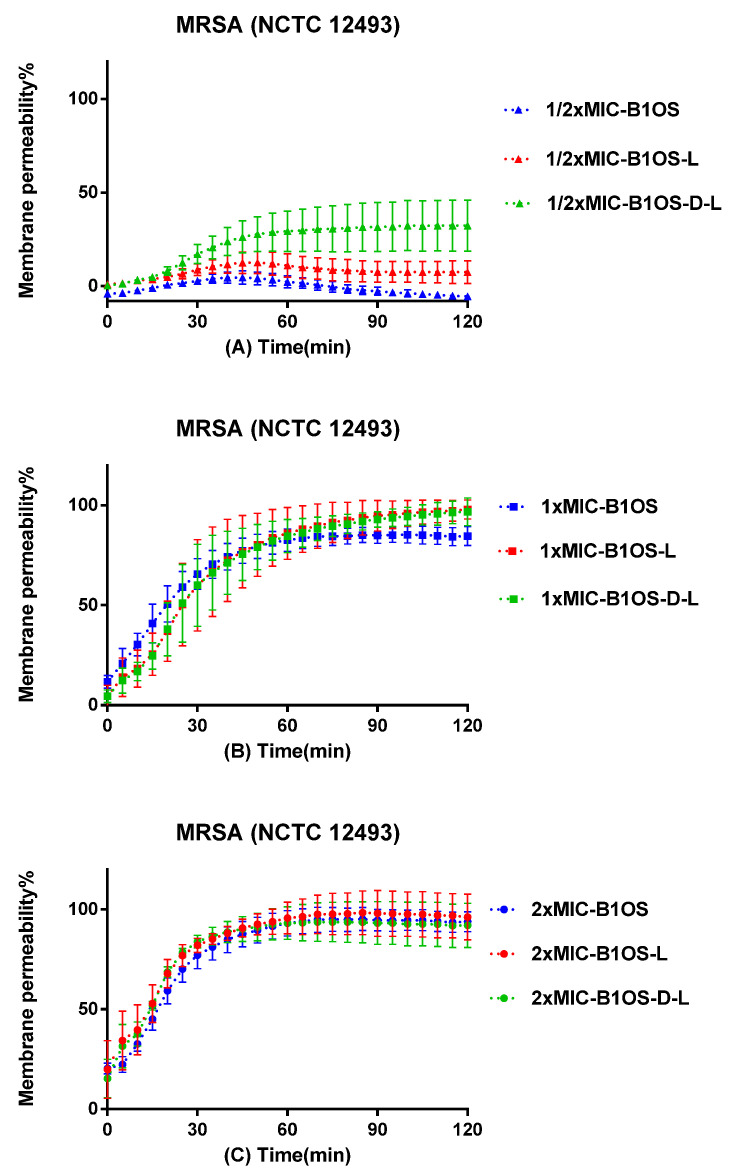
Kinetic membrane permeability curves of B1OS (blue), B10S-L (red), and B1OS-D-L (green) at the concentrations of 1/2× MIC (**A**), 1× MIC (**B**), and 2× MIC (**C**). The fluorescence was measured every 5 min. The permeability percentage was obtained by comparing the fluorescence intensity of bacterial suspension treated with 70% isopropanol. Data were represented as the means ± standard deviation of three independent experiments.

**Figure 6 toxins-13-00611-f006:**
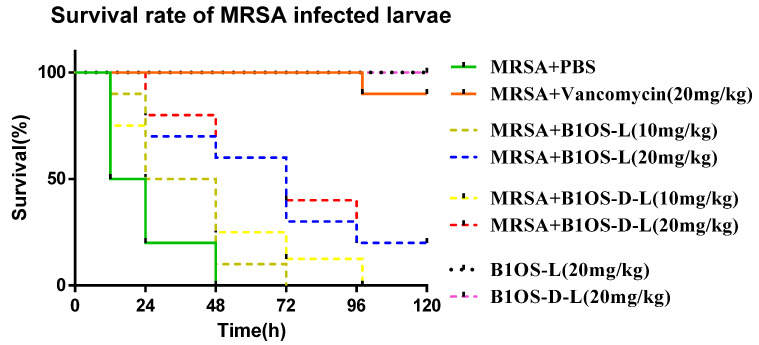
The percentage of survival in waxworms infected with MRSA. The larvae were treated with different doses of B1OS-L and B1OS-D-L (10, 20 mg/kg). Sterile phosphate-buffered saline (PBS) and 20 mg/kg of vancomycin were used as negative control and positive control. The larvae with no MRSA were treated with 20 mg/kg of B1OS-L and B1OS-D-L to evaluate the potential toxicity of peptides to the host.

**Figure 7 toxins-13-00611-f007:**
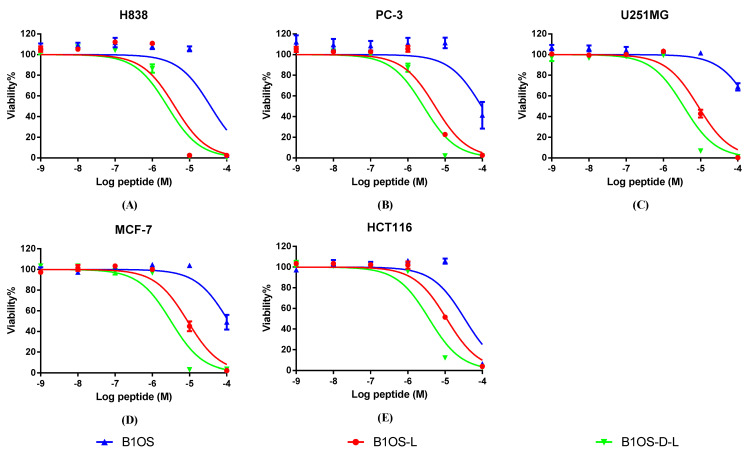
The effects of B1OS (blue), B1OS-L (red), and B1OS-D-L (green) on the proliferation of human cancer cell lines. (**A**) H838, non-small-cell lung cancer; (**B**) PC-3, human prostate carcinoma cell line; (**C**) U251MG, human glioblastoma astrocytoma; (**D**) MCF-7, human breast cancer cell line; (**E**) HCT116, human colorectal carcinoma cell line. Cells were treated with peptide for 24 h at a range of concentrations from 10^−9^ to 10^−4^ M. Cells of the positive group were treated with 0.1% Triton X-100. The curves were fitted using normalised dose–response analysis. The error bar represents the means ± standard deviation of three independent experiments. B1OS-L and B1OS-D-L showed significance against the proliferation of cancer cells at 10 μM.

**Figure 8 toxins-13-00611-f008:**
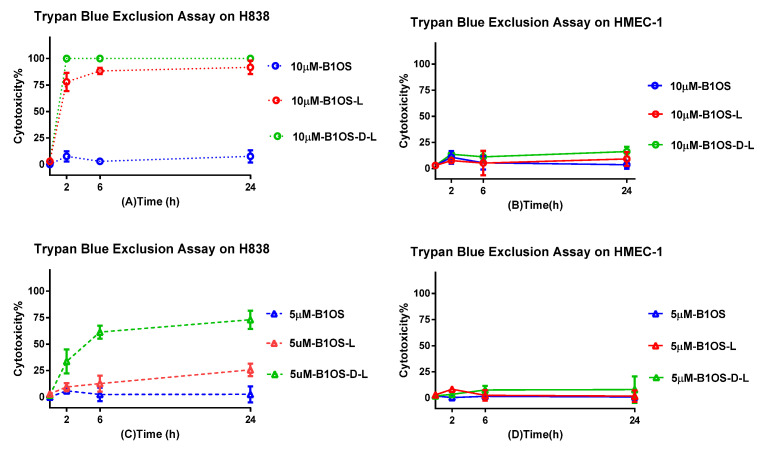
The cytotoxicity was evaluated by trypan blue exclusion assay. Human lung cancer cells H838 (**A**,**C**) and human microvascular endothelial cells HMEC-1 (**B**,**D**) were treated with peptide B1OS (blue), B1OS-L (red), and B1OS-D-L (green) at the concentration 10 μM and 5 μM for 2, 6, and 24 h, respectively. The percentage of cytotoxicity was obtained by calculating the ratio of dead cells to total cells. Data were represented as the means ± standard deviation of three independent experiments.

**Figure 9 toxins-13-00611-f009:**
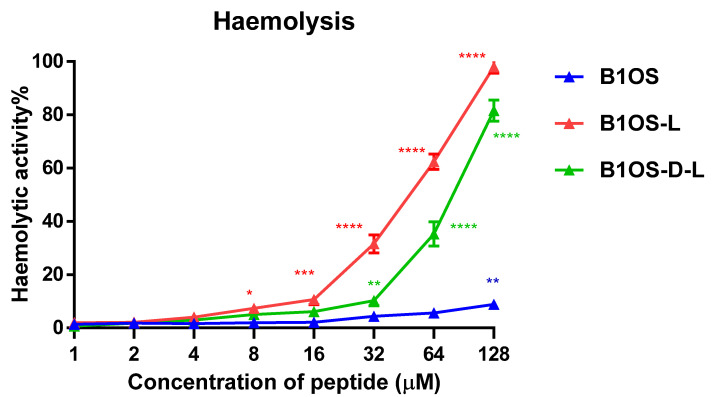
The haemolysis activity of peptides was evaluated by using horse erythrocytes with peptide concentrations from 1 μM to 128 μM. Red blood cells treated with 0.1% Triton X-100 and PBS were used as the positive control and negative control, respectively. B1OS-D-L caused less than 10% haemolysis at its maximum MIC value (32 μM). Data were represented as the means ± standard deviation of three independent experiments. The statistical significance is indicated as * (*p* < 0.05), ** (*p* < 0.01), *** (*p* < 0.001), and **** (*p* < 0.0001) versus negative control.

**Table 1 toxins-13-00611-t001:** Structural parameters of B1OS, B1OS-L, and B1OS-D-L.

Parameters.	B1OS	B1OS-L	B1OS-D-L
Hydrophobicity (H)	0.678	0.721	0.721
Hydrophobic moment (µH)	0.333	0.291	0.291
Net charge (z)	+4	+4	+4
Peptide sequence	FPLIASLAGNVVPKIFCKITKRC	FLPLIASLAGNVVPKIFCKITKRC	FLPLIASLAGNVVPKIFCKITKRC

**Table 2 toxins-13-00611-t002:** MICs and MBCs of B1OS, B1OS-L, and B1OS-D-L against six different microorganisms.

Microorganisms		MICs/MBCs (µM)			MICs (μM)
		B1OS	B1OS-L	B1OS-D-L	Norfloxacin ^a^
**Gram-positive**	*S. aureus* (NCTC 6538)	32/64	2/4	2/2	NA
	MRSA (NCTC 12493)	64/128	4/8	4/4	6.26
	*E. faecalis* (NCTC 12697)	64/128	8/8	8/8	12.53
**Gram-negative**	*Escherichia coli* (*E. coli*) (ATCC 8739)	32/32	16/32	16/16	NA
	*Klebsiella pneumonia* (*K. pneumoniae*) (ATCC 43816)	64/128	64/64	32/32	6.26
	*Pseudomonas aeruginosa* (*P. aeruginosa*) (ATCC 9027)	128/256	64/128	32/64	NA
**HC_50_ ^b^ (µM)**		739.92	29.92	74.5	NA

^a^ From [17]; ^b^ HC_50_ is the concentration of the tested peptides that caused 50% haemolysis of the horse red blood cells; NA: Not applicable.

**Table 3 toxins-13-00611-t003:** MBICs and MBECs of B1OS, B1OS-L, and B1OS-D-L against six different microorganisms.

Microorganisms		MBICs/MBECs (µM)			MBIC/MBEC (µg/mL)
		B1OS	B1OS-L	B1OS-D-L	Ciprofloxacin ^a^
**Gram-positive**	*S. aureus* (NCTC 6538)	64/ > 512	4/128	4/128	1/ > 64
	MRSA (NCTC 12493)	128/ > 512	8/128	4/128	>64/ > 64
	*E. faecalis* (NCTC 12697)	256/ > 512	16/ > 512	8/ > 512	NA
**Gram-negative**	*E. coli* (ATCC 8739)	128/512	64/256	32/128	NA
	*K. pneumoniae* (ATCC 43816)	256/ > 512	128/ > 512	64/ > 512	NA
	*P. aeruginosa* (ATCC 9027)	>512	256/ > 512	256/ > 512	NA

^a^ Data from [18]; NA: Not applicable.

**Table 4 toxins-13-00611-t004:** The IC_50_ values of B1OS, B1OS-L, and B1OS-D-L against PC-3, H838, U251MG, MCF-7, and HCT116 cancer cells.

Cell Line		IC_50_ (µM)	
	B1OS	B1OS-L	B1OS-D-L
**H838**	36.84	3.976	2.553
**PC-3**	95.94	5.473	2.629
**U251MG**	233.0	8.629	3.492
**MCF-7**	112	8.883	3.17
**HCT116**	33.69	11.03	3.721

## Data Availability

Not applicable.

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
