# Peer review of "Modification Strategy of D-leucine Residue Addition on a Novel Peptide from Odorrana schmackeri, with Enhanced Bioactivity and In Vivo Efficacy"

_toxins, 2021, doi:10.3390/toxins13090611_

Round 1

Reviewer 1 Report

Peptides represent attractive lead structures for drug discovery and development due to their high specificity and low toxicity. The submitted manuscript describes the design, structural characterization, and biological activity of brevinin-1 type of peptide, B1OS and its analogues B1OS-L and B1OS-D-L with L-Leu and D-Leu in the second position. The parent peptide B1OS was isolated from the skin of Odorrana schmackeri.  However, the reported study is an extension of existing publications on brevinin-type of peptides and their biological activities and do not contributes to significant improvement in understanding of the SAR of this class of peptides and their mode of action. In addition, the practical purpose of optimization of the dual antibacterial and anticancer activities of B1OS peptides is not clear. Typically, selectivity toward prokaryotic over eukaryotic cells is highly desirable for further optimization and development of the lead antibacterial peptide (to minimize nonspecific toxicity).  In addition, the data shown in Figure 9 do not support the statement that peptides B1OS-L and B1OS-D-L show low hemolytic activity. The technical quality of the reported study is modest.

My additional comments are the following:

  • Page 5, line 11. Replace “optimized” with “improved.”
  • Bacteria gene names are always written in italics.  Please correct throughout the text.
  • Page 5, Table 2. Unit for MIC/MBC is missing
  • Page 5, Table 3. Unit for MBIC/MBEC is missing.
  • Page 5, table 2 and 3. MIC/MBC and MBIC/MBEC for positive control (e.g. vancomycin for G+ and polymyxin for G- bacteria) should be listed as well.
  • There are two Table 3. Page 5 and page 8.
  • Page 5, line 135. “Regarding 2x MIC” replace with “At 2 x MIC”
  • Page 5, line 138. Replace “completely at 90 min” with “completely after 90 min”.
  • Page 5, line 141. “population of E. coli fell off” replace with “population of E. coli decreased”. Also, the reduction of bacterial colonies should be reported and discussed in quantitative forms (reduction expressed in CFU/mL).
  • Page 7, line 166. Replace “antibiotic” with “vancomycin”. It is not clear what the authors are attempting to demonstrate by comparison of the survival rate of MRSA infected larvae for B1OS peptides and vancomycin ? In addition, the vancomycin dose used in the experiment with wax worms should be the same as doses of B1OS-L and B1OS-D-L peptides (20 mg/kg).
  • Page 7, line 168. “Additionally, no death of larvae was observed up to a dose of 20 mg/kg.” It is not clear what this means? The data that support this statement are missing.
  • Page 7 line 174. Replace “regarded” with “used”.
  • Page 10, line 217. How statistically significant is the value of 10% hemolysis of B1OS-L at its MIC of 32 microM?
  • Page 11, lines 244-246. There are no data reported in this manuscript that support the statement that B1OS peptides show low hemolytic activity. B1OS-L and B1OS-D-L peptides are hemolytic even at their max. MIC of 32 microM.
  • Page 11. Line 252. Replace “D-type” with ‘D-Leu”.
  • Page 11. Line 274. “Therefore, the optimized antibacterial and anticancer activity could be maintained with appropriately decreased hemolysis.” It is hard to understand what this sentence means. Also, cell selectivity is important for the development of new antibiotics to minimize possibility of nonspecific toxicity. Thus, from the drug development point of view it is not clear what will be achieved by “the optimized antibacterial and anticancer activity” of B1OS type of peptides?
  • Page 11, lines 285-288. Please rephrase the whole paragraph for clarity. What is “the optimal hydrophobicity window’? What this means? How you determine this window?
  • Page 11, line 289. Typically, “enhanced antibacterial and anti-cancer activity” for peptide suggest a high possibility for nonspecific toxicity of this peptide that can severely hampered its further development as antibacterial drug.
  • Page 12, line 297. Replace “material” with “membrane”.
  • Page 12, line 311-312. The statement “which can be attributed to their poor stability” is of concern. If the tested peptides exhibit poor stability under the applied experimental conditions, this raises the question of the reliability of the obtained experimental data.
  • Page 12, lines 330. “”…behaved with comparable functions to B1OS-L with lower hemolytic activity”. It is not clear what this sentence means.
  • Page 13, line 344. Typically, liquid nitrogen is used to freeze aqueous sample following lyophilization. It is not clear how would be possible to lyophilize aqueous sample in liquid nitrogen, as suggested.
  • Page 13. Peptide synthesis. Type of the resin and resin substitution level should be reported.
  • Page 13. Peptide synthesis. 20% piperidine is not a solvent. 20% piperidine is a solution used to remove Fmoc protecting group during solid-phase peptide synthesis.
  • Page 13. Peptide synthesis. The cleavage solution is used to cleave peptide from the solid support in addition to removal of the side-chain protecting groups.
  • Page 13. Peptide synthesis. Cold diethyl ether was added to precipitated peptide from the cleavage solution.
  • Page 13. Peptide synthesis. Solution that contains 80% acetonitrile is impossible to freeze and lyophilize due to high content od acetonitrile.
  • Page 14. MIC/MIBC assays. As par the standard protocols for MIC/MIBC assays optical density at 600 nm (OD600)is measured. The statement that absorbance was measured in these assays is incorrect. Several studies concluded that OD600 measurements are very reliable and reproducible. However, that may not be the case for OD measurements at 550 nm used in the reported study.
  • Page 16, Hemolysis assays. Wavelength used to measure absorbance in hemolysis assays should be reported. Typically, the wavelength of 490 nm is used in this assay to measure haemoglobin absorbance.

Unfortunately, due to some major flows in the reported studies I cannot recommend this manuscript for publication.

Author Response

We thank the reviewer for the constructive comments and suggestions. Brevinins are a well-characterised antimicrobial peptide (AMP) family found in frog skin, and there are over 200 natural brevinin-1 peptides deposited in Uniport database so far. However, their applications are limited by high cytotoxicity. Our manuscript implies the substitution of the D-leucine residue could help address this challenge. On the other hand, the fast-killing pattern of brevinin peptides on the lung cancer cells is rare described. Furthermore, “FLP” moiety at the N-terminus of brevinine-type of peptides is highly conserved. Our work here revealed the significant role of the leucine residue at the second position, which increases the helical content, somehow contributing to the structure-activity relationship study.

Below are point-by-point response to the reviewer’s comments.

  • Page 5, line 11. Replace “optimized” with “improved.”
  • Response: We changed “optimized” to “improved” in the revised manuscript on Page 5 Line 115.
  • Bacteria gene names are always written in italics.  Please correct throughout the text.
  • Response: Changes have been made throughout the revised manuscript.
  • Page 5, Table 2. Unit for MIC/MBC is missing
  • Response: We added the concentration unit for MIC/MBC in Table 2.
  • Page 5, Table 3. Unit for MBIC/MBEC is missing.
  • Response: The concentration unit for MIC/MBC in Table 3 was added.
  • Page 5, table 2 and 3. MIC/MBC and MBIC/MBEC for positive control (e.g. vancomycin for G+ and polymyxin for G- bacteria) should be listed as well.
  • Response: The MICs of norfloxacin were added in Table 2, and the MBICs/MBECs of ciprofloxacin were added in Table 3.
  • There are two Table 3. Page 5 and page 8.
  • Response: We renumbered the tables in the correct order, and Table 4 was cited in the text.
  • Page 5, line 135. “Regarding 2x MIC” replace with “At 2 x MIC”
  • Response: We changed “Regarding” to “”At” in the revised manuscript.
  • Page 5, line 138. Replace “completely at 90 min” with “completely after 90 min”.
  • Response: Change was made accordingly.
  • Page 5, line 141. “population of E. coli fell off” replace with “population of E. coli decreased”. Also, the reduction of bacterial colonies should be reported and discussed in quantitative forms (reduction expressed in CFU/mL).
  • Response: The paragraph has been revised according to the suggestion in latest version.
  • Page 7, line 166. Replace “antibiotic” with “vancomycin”. It is not clear what the authors are attempting to demonstrate by comparison of the survival rate of MRSA infected larvae for B1OS peptides and vancomycin ? In addition, the vancomycin dose used in the experiment with wax worms should be the same as doses of B1OS-L and B1OS-D-L peptides (20 mg/kg).
  • Response: We used different concentrations of vancomycin (both 20 mg/kg and 50 mg/kg) as a positive control, and the data (20 mg/kg of vancomycin) was added in Figure 6.
  • Page 7, line 168. “Additionally, no death of larvae was observed up to a dose of 20 mg/kg.” It is not clear what this means? The data that support this statement are missing.
  • Response: "no death of larvae was observed up to a dose of 20 mg/kg.” means B1OS-L and B1OS-D-L are nontoxic to uninfected waxworms at the dose of 20mg/kg in which the survival rate was 100%. The curves of B1OS-L and B10S-D-L were overlapped in our original manuscript. We changed the pattern of the curve of B1OS-L, and now it can be clearly seen in Figure 9.
  • Page 7 line 174. Replace “regarded” with “used”.
  • Response: We changed “regard” to “used” in the revised version.
  • Page 10, line 217. How statistically significant is the value of 10% hemolysis of B1OS-L at its MIC of 32 microM?
  • Response: The statistical significance was added in the revised version. The statistically significant value of 10% hemolysis of B1OS-D-L at 32 uM is ** (p < 0.01).
  • Page 11, lines 244-246. There are no data reported in this manuscript that support the statement that B1OS peptides show low hemolytic activity. B1OS-L and B1OS-D-L peptides are hemolytic even at their max. MIC of 32 microM.
  • Response: We compared the HC50 values of these three peptides, and these data were added in Table 2. Yes, we observed that B1OS-L and B1OS-D-L peptides are hemolytic at their max. MIC of 32 µM.The MIC of B1OS-D-L against Gram-positive bacterial ranged from 2 µM to 8 µM in which no significant haemolysis was observed, while peptide B1OS-L demonstrated significant haemolysis from a concentration of ≥ 8 µM. These support the statement that B1OS-D-L peptide show lower hemolytic activity than B1OS-L. On the other hand, peptides from brevinin-1 family generally exhibit strong haemolysis activity. B1OS-D-L performed comparable or even better antibacterial activity and lower haemolysis activity than other brevinin-1 family peptides (e.g., Ose [1], brevinin-1H [2] and B1A [3].
  • Page 11. Line 252. Replace “D-type” with ‘D-Leu”.
  • Response: We changed “D-type” to “used” in the revised version.
  • Page 11. Line 274. “Therefore, the optimized antibacterial and anticancer activity could be maintained with appropriately decreased hemolysis.” It is hard to understand what this sentence means. Also, cell selectivity is important for the development of new antibiotics to minimize possibility of nonspecific toxicity. Thus, from the drug development point of view it is not clear what will be achieved by “the optimized antibacterial and anticancer activity” of B1OS type of peptides?
  • Response: Thank you for raising this point. Peptides from brevinin-1 family generally exhibit strong haemolysis activity. First of all, the parent peptide B1OS behaved less potent antibacterial and anti-cancer activities, which could be enhanced via modification. After substituting L-leucine by D-leucine, the hydrophobicity of peptide was maintained and the helicity content was reduced partly. B1OS-D-L exhibited comparable antibacterial activity to B1OS-L, but the hemolysis of B1OS-D-L was lower than that of B1OS-L. On the other hand, B1OS-D-L performed comparable or even better antibacterial activity and lower haemolysis activity than other brevinin-1 family peptides (e.g., Ose [1], brevinin-1H [2] and B1A [3]. This study also pointed out that comparing with B1OS-L, the employment of D-leucine residue could reduce the haemolysis while enhancing antibacterial and anti-cancer activities. Moreover, the cytotoxicity had been evaluated on normal cells and lung cancer cells using trypan blue exclusion assay, and the results indicated that B1OS-D-L could kill the lung cancer cells without harming human normal cells at a concentration of 10 µM (Figure 8). We believe these evidences suggest B1OS-D-L has a selective toxicity in vitro.
  • So far, the number of people who are suffering from cancer and multi-resistant infections has increased globally, such that both diseases are already seen as current and future major causes of death. In addition, chronic infections are one of the leading causes of cancer, resulting from the instability in the immune system that allows cancer cells to proliferate. Antimicrobial peptides with dual antibacterial and anticancer activity are viewed as future chemotherapeutic drugs. Our finding here, therefore, is significant.
  • Page 11, lines 285-288. Please rephrase the whole paragraph for clarity. What is “the optimal hydrophobicity window’? What this means? How you determine this window?
  • Response: The optimum hydrophobicity window also called the threshold hydrophobicity window, which was proposed by Yuxin Chen and her colleagues. Researchers systematically increase or decrease the hydrophobicity of antimicrobial peptides via amino acid substitution and found that there was an optimum hydrophobicity window in which high antimicrobial activity could be obtained, decreased or increased hydrophobicity beyond this window dramatically decreased antimicrobial activity [4].
  • Page 11, line 289. Typically, “enhanced antibacterial and anti-cancer activity” for peptide suggest a high possibility for nonspecific toxicity of this peptide that can severely hampered its further development as antibacterial drug.
  • Response: This has been answered above. The cytotoxicity had been evaluated in this study, indicating that B1OS-D-L shows a fast-killing effect on lung cancer cells without harming human normal cells (HMec-1).
  • Page 12, line 297. Replace “material” with “membrane”.
  • Response: We changed “material” to “membrane” in the revised manuscript.
  • Page 12, line 311-312. The statement “which can be attributed to their poor stability” is of concern. If the tested peptides exhibit poor stability under the applied experimental conditions, this raises the question of the reliability of the obtained experimental data.
  • Thank you for raising this point. The original description aimed to raise the limitation of this study, but the inappropriate presentation may cause misunderstanding. Peptides are generally unstable under physiological salt and serum conditions and are easily degraded by proteases, resulting in poor in vivo activity. In this study, the experiments were taken in vitro excepted the waxworm model that was the in vivo analysis of antibacterial activity. Therefore, the in vitro experimental condition was applied with the absence of those salt, serum, proteases. The in vivo study of antibacterial activity was performed using waxworm model, B10S-L and B1OS-D-L could reduce the mortality rate of infected waxworm but was less potent compared with vancomycin. Herein, G. mellonella model serves as a rapid and reproducible model for assessing the antimicrobial activity of peptides against MRSA. However, G. mellonella model can not represent animal models. One of the main reasons is that it lacks an adaptive immune system, and this is the limitation of this study. Therefore, more further investigations are needed to evaluate the potency of these peptides in vivo.
  • Page 12, lines 330. “”…behaved with comparable functions to B1OS-L with lower hemolytic activity”. It is not clear what this sentence means.
  • Response: Response: After replacing L-leucine residue by D-leucine at 2nd position, B1OS-D-L behaved comparable functions to B1OS-L, moreover, the haemolysis activity of B1OS-D-L was lower than B1OS-L. The sentence has been revised in the latest manuscript.
  • Page 13, line 344. Typically, liquid nitrogen is used to freeze aqueous sample following lyophilization. It is not clear how would be possible to lyophilize aqueous sample in liquid nitrogen, as suggested.
  • Response: In our group, the aqueous sample was snap-frozen with liquid nitrogen prior to lyophilisation. It is unlikely to lyophilise aqueous sample in liquid nitrogen. This mistake has been rectified on Page 14 Line 375-376.
  • Page 13. Peptide synthesis. Type of the resin and resin substitution level should be reported.
  • Response: Thank you for raising this problem. The type of the resin and resin substitution level were added in the revised manuscript.
  • Page 13. Peptide synthesis. 20% piperidine is not a solvent. 20% piperidine is a solution used to remove Fmoc protecting group during solid-phase peptide synthesis.
  • Response: We changed “four solvents” to “four solutions” in the revised manuscript on Page 14 Line 411.
  • Page 13. Peptide synthesis. The cleavage solution is used to cleave peptide from the solid support in addition to removal of the side-chain protecting groups.
  • Response: We rephrased the purpose(s) of using cleavage solution accordingly. 
  • Page 13. Peptide synthesis. Cold diethyl ether was added to precipitated peptide from the cleavage solution.
  • Response: This information was added in the revised manuscript on Page 15 Line 417-418.
  • Page 13. Peptide synthesis. Solution that contains 80% acetonitrile is impossible to freeze and lyophilize due to high content od acetonitrile.
  • Response: We actually used 5 mL of acetonitrile/H2O/TFA (80/19.95/0.05, v/v/v) and 15 mL of H2O/TFA (99.95/0.05, v/v) to dissolve peptide before lyophilisation, which was stated in the original manuscript. The solution eventually contained 20% acetonitrile.
  • Page 14. MIC/MIBC assays. As par the standard protocols for MIC/MIBC assays optical density at 600 nm (OD600)is measured. The statement that absorbance was measured in these assays is incorrect. Several studies concluded that OD600 measurements are very reliable and reproducible. However, that may not be the case for OD measurements at 550 nm used in the reported study.
  • Response: Thank you for the suggestion. However, we disagree with reviewer on this point because the 550nm wavelength was also used in our previously published works. Hundreds of studies proved that OD measurements at 550 nm is very reliable and reproducible. We therefore would like to keep this statement.
  • Page 16, Hemolysis assays. Wavelength used to measure absorbance in hemolysis assays should be reported. Typically, the wavelength of 490 nm is used in this assay to measure haemoglobin absorbance.
  • Response: The wavelength for measuring haemoglobin absorbance was added.

Reviewer 2 Report

The article described the identification of peptide B1OS from the Odorrana schmackeri skin secretion cDNA library. The peptide was classified as a member of the Brevinin-1 peptide family due to its sequence homology. However, a leucine residue close to the N-terminus, which is conserved in most members of the Brevinin-1 family, is missing in B1OS. The authors engineered two B1OS variants, B1OS-L and B1OS-D-L, which contain the addition of a L-leucine and D-leucine residue at this site, respectively. The authors assessed the secondary structures, antimicrobial activities, anti-biofilm activity, bactericidal killing-kinetics, in vivo antibacterial activities, anti-cancer activity, cytotoxicity and haemolysis activity of B1OS and its variants. They concluded that the native B1OS peptide basically has lower activity in all the biological assays performed when compared with the leucine variants and they speculated that the reason is due to its lower helical content and hydrophobicity, which reduced its effectiveness in pore formation on cell membrane. The D-leucine variant, B1OS-D-L also seems have greater potential in developing as antimicrobial/anti-cancer agent as it has similar antibacterial and anti-cancer activity as the L-leucine variant but lower cytotoxic and haemolytic activity.

It is an impressive study with an extensive analysis on the biological functions of the peptides but I have the following queries/suggestions.

Results:

  • Figure 1: Is the DNA sequence after the stop codon (from number 212) illustrating anything important specific for this study? I do not understand why they are shown.
  • The secondary structure and hydrophobicity prediction was done using “I-TASSER” but the processed was not described in the method. I wonder how the software/algorithm worked and what the prediction was based on but the sentence was referred to a patent by Jiang et al and the title does not seem to be relevant to the method. Please provide appropriate methods and references.
  • The CD study was conducted in 50% TFE which the authors claim to imitates the membrane environment. I disagree and instead believe that TFE tends to induce the formation of helices in protein/peptides no matter how relevant it is to their native structure. Please repeat the experiment in aqueous or other solvents to show the peptides’ helical propensity.
  • Line 97: TFE was misspelled as TFA
  • Figure 2: The whole figure is of extremely low resolution and the amino acid labels in the structure images are not even illustrated properly. Please tidy up the labels and improve the digital resolution of the figure.
  • Table 1: “Hydrophobicity” misspelled in Table 1.
  • Net charge should have + or - sign
  • A description on how the hydrophobicity was measured or how the numbers were generated is required (this is related to the previous comment about method/referencing I-TASSER)
  • Pay attention to all the abbreviation used in the manuscript. E.g. CD, MIC, MBC, MBIC, MBEC….etc. They should all be spelled out when they appeared the first time. Readers are not necessarily familiar with all the abbreviations.
  • Table 2 and 3. Units are missing in both tables.
  • The antibacterial and anti biofilm assays: Dose-response (e.g. growth inhibition %) curves should be included for these studies.
  • The author can consider combining the killing kinetic result figure with the dose-response result figure to make a more compact figure.
  • Membrane permeability assay: I wonder how quickly the fluorescence signals would reach 100% for the 70% isopropanol (positive) control in this assay. Is that instant or does it take some time too?
  • The author mentioned that the cell permeability was affected by the peptides in a “concentration-dependent manner” but all they showed was 1 concentration (2xMIC) over time. How does it show the concentration dependency?
  • The results of this assay is quite interesting but I wonder if really provides strong evidence for membrane destruction or pore formation of the peptide. Is it possible that the membrane just became more permeable without forming pores or being completely disrupted? A negative control with just the dye but no peptide will also be helpful in proving that the dye does not just diffuse into the cell via another other mechanisms.
  • The cytotoxicity analysis should again be presented in a dose-response format instead of showing only 2 concentrations, especially when the authors is claiming that it is “dose-dependent”. I appreciate the importance of showing the cytotoxicity effect over time though.
  • Figure 9: The micromolar symbol is not displaying properly on the X-axis

Discussion:

  • The authors discussed the potential evolution process of the peptides and suggested that the N-terminal FLP motif was evolved originally from the FP motif. I am not an expert in molecular evolution but I wonder if the authors can support their claim/hypothesis by doing a phylogenetic analysis of these sequences? Afterall it is quite surprising that the B1OS native peptide is expressed despite its poor activity. In fact, do we know if this peptide is actually expressed (by e.g. proteomic analysis)? Or if the transcript count is somewhat abundance as compared to other peptides? And in the transcriptome, did you also find other Brevinin 1 members, esp. the one with the leucine in the N-terminus?
  • How does the activity of these B1OS-L or B1OS-D-L compare to the reported activity of the other Brevinin-1 family members? Or how does it compared to some general antimicrobials drugs in the market? It is difficult for the readers to know what’s a real “promising” activity is unless we have something to compare to.
  • Paragraph 6 in Discussion. I don’t understand how the concentration dependency can explain how the peptide interacts with the lipid membrane, i.e. whether it is consistent with the carpet model mechanism. Please clarify this speculation.
  • Paragraph 7 in Discussion: The authors said the lower potency of the B1OS-L/DL peptides compared to the current antibiotics might be attributed to their lower stability. Is it because they are peptides and peptides are generally less stable in vivo? Or is it something specifically to B1OS and their variants? It’d be nice if you could clarify.

Methods:

  • HPLC chromatograms and MS spectra should be provided as supplementary material to show the purity of the samples.
  • As suggested above, secondary structure prediction work is not described in the Method section.
  • In the MIC/MBC assays, Line 402, what was the peptide stock dissolved in? Plain water?

Author Response

We thank the reviewer for the overall positive evaluation of our work. 

Below are our point-by-point response to the reviewer’s comments.

  • Figure 1: Is the DNA sequence after the stop codon (from number 212) illustrating anything important specific for this study? I do not understand why they are shown.
  • Response: The nucleotide sequence after the stop codon in Figure 1 was deleted.
  • The secondary structure and hydrophobicity prediction was done using “I-TASSER” but the processed was not described in the method. I wonder how the software/algorithm worked and what the prediction was based on but the sentence was referred to a patent by Jiang et al and the title does not seem to be relevant to the method. Please provide appropriate methods and references.
  • Response: Thank you for noticing this issue. This mistake was recruited in Methods and Materials section on Page 16. 
  • The CD study was conducted in 50% TFE which the authors claim to imitates the membrane environment. I disagree and instead believe that TFE tends to induce the formation of helices in protein/peptides no matter how relevant it is to their native structure. Please repeat the experiment in aqueous or other solvents to show the peptides’ helical propensity.
  • Response: We deleted this statement from the manuscript, and the CD spectra of B1OS and its analogues in aqueous solvent was added.
  • Line 97: TFE was misspelled as TFA
  • Response: This problem was corrected.
  • Figure 2: The whole figure is of extremely low resolution and the amino acid labels in the structure images are not even illustrated properly. Please tidy up the labels and improve the digital resolution of the figure.
  • Response:  These problems were fixed, and a proper figure legend was added.
  • Table 1: “Hydrophobicity” misspelled in Table 1.
  • Response: This problem was corrected in Table 1.
  • Net charge should have + or – sign
  • Response: The net charge sign was added in Table 1.
  • A description on how the hydrophobicity was measured or how the numbers were generated is required (this is related to the previous comment about method/referencing I-TASSER)
  • Response: This mistake was recruited in Methods and Materials section on Page 16.
  • Pay attention to all the abbreviation used in the manuscript. E.g. CD, MIC, MBC, MBIC, MBEC….etc. They should all be spelled out when they appeared the first time. Readers are not necessarily familiar with all the abbreviations.
  • Response: These mistakes have been rectified.
  • Table 2 and 3. Units are missing in both tables.
  • Response: The concentration units in Table 2 & 3 were added.
  • The antibacterial and anti biofilm assays: Dose-response (e.g. growth inhibition %) curves should be included for these studies.
  • Response: The graphs of viability% and biofilm mass% (referred to Figure S3 and S4) were added in the supplementary material document.
  • The author can consider combining the killing kinetic result figure with the dose-response result figure to make a more compact figure.
  • Response: Thanks for suggestion, but it would be too crowded to put all figures in one. Therefore, we separated the killing kinetic result figure and dose-response result figure. The dose-response result figure (Figure S2) was added in the supplementary material document.
  • Membrane permeability assay: I wonder how quickly the fluorescence signals would reach 100% for the 70% isopropanol (positive) control in this assay. Is that instant or does it take some time too?
  • Response: Response: When preparing the positive control, the bacteria was fully vortexed before adding the 70% isopropanol, and then placed in the room temperature for one hour to ensure that the bacterial membrane is destroyed. The fluorescence signal of 100% membrane destruction will be obtained immediately after adding the dye.
  • The author mentioned that the cell permeability was affected by the peptides in a “concentration-dependent manner” but all they showed was 1 concentration (2xMIC) over time. How does it show the concentration dependency?
  • Response: Kinetic membrane permeability curves at the concentration of 1/2 x MIC and 1 x MIC were added in the latest version to support this statement.
  • The results of this assay is quite interesting but I wonder if really provides strong evidence for membrane destruction or pore formation of the peptide. Is it possible that the membrane just became more permeable without forming pores or being completely disrupted? A negative control with just the dye but no peptide will also be helpful in proving that the dye does not just diffuse into the cell via another other mechanisms.
  • Response: Yes, it’s possible that the bacterial membrane just became more permeable without forming pores or being completely disrupted Certainly, we performed the negative control in which the bacterial suspension was treated without peptides, and its fluorescence signal remained almost unchanged which indicated that the membrane would not become permeable. This also means that the dye could not diffuse into the bacterial membrane unless the membrane is destroyed by peptide. As shown in the revised kinetic permeability figure, peptide failed to permeabilised bacterial membrane at low concentration (1/2 x MIC).
  • The cytotoxicity analysis should again be presented in a dose-response format instead of showing only 2 concentrations, especially when the authors is claiming that it is “dose-dependent”. I appreciate the importance of showing the cytotoxicity effect over time though.
  • Response: Thank you for raising this point. We agreed with the reviewer that the expression of “dose-dependent” is improper.  This statement was removed from the revised version.
  • Figure 9: The micromolar symbol is not displaying properly on the X-axis
  • Response: This problem was fixed in the latest version.

Discussion:

  • The authors discussed the potential evolution process of the peptides and suggested that the N-terminal FLP motif was evolved originally from the FP motif. I am not an expert in molecular evolution but I wonder if the authors can support their claim/hypothesis by doing a phylogenetic analysis of these sequences? Afterall it is quite surprising that the B1OS native peptide is expressed despite its poor activity. In fact, do we know if this peptide is actually expressed (by e.g. proteomic analysis)? Or if the transcript count is somewhat abundance as compared to other peptides? And in the transcriptome, did you also find other Brevinin 1 members, esp. the one with the leucine in the N-terminus?
  • Response: Thank you for raising this point. We also agree that making a phylogenetic analysis will help to illustrate this hypothesis. However, we are not the expert in biology, and we don’t know how to make a phylogenetic analysis. Yes, B1OS is actually expressed, the RP-HPLC chromatogram of the skin secretion of Odorrana schmackeri was added as a supplementary (Figure S1). The sequence of B1OS was further determined by MS/MS sequencing in the latest manuscript. A brevinin 1-type peptide with one leucine in the N-terminus was also found in skin secretion of Odorrana schmackeri, which has been described in our previous study reported by Xiaowei Zhou [1].
  •  
  • How does the activity of these B1OS-L or B1OS-D-L compare to the reported activity of the other Brevinin-1 family members? Or how does it compared to some general antimicrobials drugs in the market? It is difficult for the readers to know what’s a real “promising” activity is unless we have something to compare to.
  • Response: The description of the comparison of B1OS-D-L with other brevinin-1 peptide was added in the revised manuscript on Page 13 Line 333-338.
  •  
  • Paragraph 6 in Discussion. I don’t understand how the concentration dependency can explain how the peptide interacts with the lipid membrane, i.e. whether it is consistent with the carpet model mechanism. Please clarify this speculation.
  • Response: Combining the results of time-killing and membrane permeability, it can be found that peptides killed bacteria in a concentration-dependent pattern. We speculated that 1 x MIC and 2 x MIC of peptides might act in different mechanisms. Hartmann et al. used the SEM, and TEM images proposed that AMPs only induce membrane leakage at a low concentration, while higher concentrations (> MIC) could cause membrane disintegration and cell lysis [5]. Researchers also proposed that 1xMIC of PSN-PC could not cover the entire lipid layer; however, peptides were proposed to tended to act in a carpet and toroidal pore mode under the concentration of 2 x MIC and 4 x MIC [6]. In this case, B1OS, B1OS-L and B1OS-D-L were likely to act in the same way. The paragraph has been revised in the latest version on Page 14.

  • Paragraph 7 in Discussion: The authors said the lower potency of the B1OS-L/DL peptides compared to the current antibiotics might be attributed to their lower stability. Is it because they are peptides and peptides are generally less stable in vivo? Or is it something specifically to B1OS and their variants? It’d be nice if you could clarify.
  • Response: The sentence has been revised in the latest manuscript on Page 14 Line 341-345.

Methods:

  • HPLC chromatograms and MS spectra should be provided as supplementary material to show the purity of the samples.
  • Response: The HPLC chromatograms and MALDI-TOF spectra of purified peptides were added in the supplementary (Figure S5 and S6).
  • As suggested above, secondary structure prediction work is not described in the Method section.
  • Response: Change was made according to reviewer’s suggestion.
  • In the MIC/MBC assays, Line 402, what was the peptide stock dissolved in? Plain water?
  • Response: The peptide was dissolved in DMSO as the peptide stock solution. This information was added in the revised version on Page 15 Line 451.

Round 2

Reviewer 1 Report

Although the revised manuscript is improved, better justification for the development of antimicrobial peptides with dual antibacterial and anticancer activity is still needed.

Author Response

We thank you again for referees' comments. As suggested by reviewer, we have added the relative description of the motivation for developing dual active peptides in the Introduction section in the latest manuscript (on Page 1 Line 32-38, Line 44-47).

Reviewer 2 Report

I appreciate most of the responses and changes from the authors. However with the new CD spectra provided on the peptide in aqueous buffer, it is clear that the peptide does not adopt a helical conformation in the aqueous solution, and instead it is random coil. The authors did not discuss this discrepancy with the TFE result, instead they modified the sentence to:

"It indicated that B1OS-L tends to form a higher alpha-helix structure than B1OS in 50% trifluoroethanol (TFE)/10 mM ammonium acetate (NH4AC) solution."

This sentence is misleading because the readers may think it means that the peptide adopts helical structure in either TFE or NH4Ac, which is incorrect. This result also suggests that in all the biological assays conducted, where no TFE is present, the peptide conformed to random coil rather than alpha helix. This might not be a problem as the mechanism of action of this peptide is unexplored anyway but this needs to be clarified and discussed in the manuscript. It is disappointing to see that the authors basically avoided it. 

Author Response

Thank you again for your constructive comments and suggestions. We apology for the inappropriate and insufficient description causing a misunderstanding. These sentences have been corrected (the changes can be found in the latest version Page 3, Line 105-110; Page 11, Line 268). We also added a relative discussion on page 12, paragraph 2 in the Discussion section.

Round 3

Reviewer 2 Report

I am happy to see the changes and I see no further critical issues.